# Glycemia after Betamethasone in Pregnant Women without Diabetes—Impact of Marginal Values in the 75-g OGTT

**DOI:** 10.3390/healthcare8010040

**Published:** 2020-02-17

**Authors:** Ioannis Kakoulidis, Ioannis Ilias, Anastasia Linardi, Aikaterini Michou, Charalampos Milionis, Foteini Petychaki, Evangelia Venaki, Eftychia Koukkou

**Affiliations:** Department of Endocrinology, Diabetes and Metabolism, Elena Venizelou General and Maternity Hospital, GR-11521 Athens, Greece; i_kakoulidis@yahoo.gr (I.K.); an.linardi@yahoo.gr (A.L.); katerina.michoy@yahoo.com (A.M.); pesscharis@hotmail.com (C.M.); petychakifoteini@yahoo.com (F.P.); evivenaki@gmail.com (E.V.); ekoukkou@gmail.com (E.K.)

**Keywords:** betamethasone, pregnancy, glycemia, insulin, oral glucose test

## Abstract

Betamethasone (BM) administration in pregnancy has been shown to reduce the incidence and severity of neonatal respiratory distress syndrome. Its known diabetogenic impact, combined with placental insulin resistance, leads to a transient increase in glycemia. However, its effect on glucose homeostasis in pregnancy has not been adequately investigated. We closely monitored and assessed the glycemic profile of 83 pregnant women, with normal glucose metabolism, who were given BM during their hospitalization due to threatened premature labor. A significant change in the glycemic profile in most patients was noted, lasting 1.34 ± 1.05 days. Sixty-six of eighty-three women were eventually treated with insulin to maintain glycemia within acceptable limits. The mean ± SD insulin dosage was 12.25 ± 11.28 units/day. The need for insulin therapy was associated with higher BM doses and the presence of marginal values in the 75-g oral glucose tolerance test (OGTT) at 60 min. Our study demonstrates, following BM administration, the need for increased awareness and individualized monitoring/treatment of pregnant women with normal—yet marginal—values in the 75-g OGTT.

## 1. Introduction

The administration of corticosteroids (mainly betamethasone, BM) in pregnancy is common practice in obstetrics, with the aiming of reducing the incidence/severity of respiratory distress syndrome and of intraventricular hemorrhage in neonates. Usually one or two doses of BM, separated by 24 h, are administered intramuscularly mainly—but not exclusively—from the 23rd to 34th week of gestation, with the aim of promoting the maturation of the lungs of the embryo, particularly when preeclampsia or premature rupture of membranes may lead to premature birth (or imminent birth within 7 days) [1,2,3,4,5,6]. 

The diabetogenic potential of BM is known, and combined with placental insulin resistance in pregnancy, leads to a transient increase in the blood glucose levels of pregnant women [7,8]. Few small studies in the literature exist regarding the action of BM on glucose homeostasis in pregnant women. With small numbers of subjects, they may lack statistically reliable conclusions [9]. Great attention is paid to the group of pregnant women who have either pre-existing diabetes or gestational diabetes, and especially those under insulin therapy. In this case, it is recommended, by limited bibliographic data, to up-titrate the insulin dose by 30%–40% in order to prevent very high blood glucose levels, which may exacerbate an already at-risk pregnancy (hydramnios) [7,8,9,10,11,12,13,14,15,16]. Furthermore, hyperglycemia can lead to severe neonatal hypoglycemia in the case of preterm delivery [7,9,12]. More recent studies, also with a relatively small number of subjects, showed that this increase in insulin dose may be inadequate [7,8,10,12,16]. Additionally, there has been growing emphasis in the literature on women with gestational diabetes under medical nutrition therapy or, in some cases, with apparent normal glucose metabolism, where—nevertheless—hyperglycemia may occur, necessitating the initiation of transient insulin therapy [8,10,17]. For this reason, some researchers recommend intensive monitoring of pregnant women during the time of BM administration [8,9,11]. 

Based on the above, the purpose of this work was to study changes in the glycemic profile of women with singleton pregnancy and normal glucose metabolism, after administration of BM at 12 mg/day for one or two days, during their hospitalization for at-risk pregnancy, and to study the possible factors that are associated with the need for insulin.

## 2. Materials and Methods

We conducted an observational study monitoring the glycemic profile of 83 Greek (Caucasian) women, with singleton pregnancy (mean age ± SD: 32.0 ± 4.7 years), during the 31.3 ± 3.7th week of gestation (59 women <34th week; 24 women >34th week), who received BM during their hospitalization for at-risk pregnancy (hydramnios, premature rupture of membranes, uterine contractions, vaginal bleeding), from August 2016 to December 2018. All women included in the study had a normal—for pregnancy—glucose metabolism profile. The latter was presumed after a normal oral glucose tolerance test (OGTT) with 75 g of glucose loading, performed at 24–28 weeks of gestation, or in the case of pregnancies before 24 weeks, with morning fasting blood glucose within normal limits for pregnancy (<92 mg/dL) [18]. Women with past medical history of diabetes, gestational diabetes, impaired fasting glucose, impaired glucose tolerance or metabolic syndrome were excluded. Sixty-six women received a total of 24 mg of BM and 17 women received 12 mg of BM. Thirteen women (15.6%) had hydramnios in obstetric ultrasound and forty-five women (54.2%) had thyroid disease (21 with known hypothyroidism before pregnancy under treatment, 20 with hypothyroidism that was diagnosed in pregnancy who received treatment upon diagnosis, 3 with subclinical hyperthyroidism that did not receive treatment, and 1 that had thyroidectomy due to nodular thyroid disease).

The mean weight gain in pregnancy was 10.1 ± 4.9 kg. Monitoring of the women’s glycemic profile was based on six to seven capillary blood glucose measurements per day (pre and one-hour postprandial, plus an overnight measurement, if possible) with point of care devices (POC). The timing of the glucose monitoring was based on previously published research, which has shown that, following each dose of betamethasone, plasma glucose levels are expected to gradually increase, with a peak of hyperglycemia at 8–10 h [7,12]. All patients were under a balanced dietary regimen during their hospitalization. The intervention and correction of hyperglycemia was carried out in accordance with international guidelines for pregnancy, with target glucose levels of 90 to 140 mg/dL (fasting and one-hour postprandially, respectively) [11,18,19]. Age, gestational age, weight change in pregnancy, days of hyperglycemia, the presence of hydramnios, the need for insulin initiation, the total daily insulin dosage after BM, glucose levels in the oral glucose tolerance test (OGTT) with 75 g of glucose loading (66/83 women had done the OGTT), as well as the coexistence of thyroid disease, were recorded. Stratification of the OGTT results was done per the glycemic ranges provided by the HAPO trial [20]. Statistical analysis was done with stepwise backward logistic regression. The study was approved by our hospital’s scientific board/ethics committee.

## 3. Results

There was considerable change in the glycemic profile of most women, mainly in the first 24 hours of BM administration, which lasted on average 1.34 ± 1.05 days, with maximum values of fasting capillary blood glucose at 119.6 ± 22.3 mg/dL, premeal at 127.9 ± 23.6 mg/dL and one-hour postprandial at 162.6 ± 25.2 mg/dL. In 66 out of 83 pregnant women (79.5%), insulin had to be administered after one or two doses of BM, targeting glycemia between 90 and 140 mg/dL (fasting and one-hour postprandially, respectively; see the flowchart in Figure 1). 

In this group of pregnant women, hyperglycemia lasted on average 1.69 ± 0.89 days and 12.25 ± 11.28 insulin units/day were administered, resulting in fasting glucose levels of 121.9 ± 21.9 mg/dL, 130.9 ± 22.4 mg/dL premeal, and 165.2 ± 24.6 mg/dL one-hour postprandially. In about half of these women (29/66–43.9%), it was considered necessary to use an intensive regimen with a combination of basic and mealtime insulin. No hypoglycemic events were noted in any of these women at any time during insulin administration.

The need for insulin administration showed an association with the higher/two-day BM dosage (*p* = 0.013) but not with maternal age, gestational age, body weight change, presence of hydramnios, or presence of thyroid disease. In 60/83 women with OGTT, mean glycemia was 79.6 ± 5.9 mg/dL, 126.0 ± 23.5 mg/dL, and 102.0 ± 21.0 mg/dL at 0, 60, and 120 min of the OGTT, respectively. Of these OGTTs, 28 were considered to be marginal (at the upper limit of normal and at least 80 mg/dL; HAPO category 3 and higher) at 0 min, 25 were considered to be marginal (at the upper limit of normal and at least 133 mg/dL; HAPO category 3 and higher) at 60 min, and 23 were considered to be marginal (at the upper limit of normal and at least 126 mg/dL; HAPO category 3 and higher) at 120 min (Figure 2). 

When these marginal values were also considered in the analysis, a statistically significant association (*p* = 0.050) was noted for the need for insulin administration with the presence of marginal values at 60 min of the OGTT; there was no association with the presence of marginal values at 0 min or at 120 min (*p* > 0.50).

## 4. Discussion

As expected, a change in the glycemic profile following BM was noted [7,8,12,15,17], with a large proportion of pregnant women (79.5%) requiring insulin administration for a short period of time. This was associated with the dosing of BM but not with other key parameters such as gestational age or weight gain in pregnancy. Even the presence of hydramnios, which could have been related to a possible mild disorder of glucose homeostasis not detected with OGTT, was not associated with the need for insulin. However, the proportion of pregnant women with hydramnios in our study was relatively small (15.6%), thus limiting the ability to reach more robust conclusions. A possible link of the need for insulin administration with the presence of thyroid disease (in the context of increased risk for gestational diabetes with coexisting hypothyroidism [21]) was also not observed. 

The definition and stratification of marginal blood glucose levels in the OGTT corresponded to the classification of the glycemic profile in the HAPO study [20]. The presence of a statistically significant association between the need for insulin administration following BM and the presence of marginal blood glucose levels at 60 min of OGTT is a possible indication of a mild glucose homeostasis disorder. This tentative disorder, although undetected during screening for gestational diabetes with OGTT and fasting blood glycemia, was possibly revealed by the administration of BM.

By design there were limitations in the study regarding the heterogeneity and variation of the factors and conditions leading to an at-risk pregnancy. Furthermore, we utilized POC devices rather than continuous glucose monitoring (CGM), which could have been more accurate in glucose profiling. However, the need for transient insulin administration in order to control glycemia after administration of BM in a large proportion of pregnant women with high-risk pregnancies remains particularly important. This is consistent with a transient increase in glycemia in women with no known gestational diabetes reported in a small number of published articles in the literature [5,8,10,12]. Having a relatively large number of subjects, this study indicates the necessity for closer monitoring of the glycemic profile of women with at-risk pregnancy, during the administration of BM, even when they have a previously normal glycemic profile, especially when there is presence of marginal values in the 75-g OGTT. However, further relevant studies, with larger and more homogenized statistical patient samples, are needed to achieve reliable conclusions.

## 5. Conclusions

In our study of pregnant women, the need for insulin therapy was associated with higher BM doses and the presence of marginal values in the 75-g oral glucose tolerance test (OGTT) at 60 min. Considering these results, our study demonstrates the need for increased awareness and individualized monitoring and treatment of pregnant women, following BM administration, with normal—yet marginal—values for the 75-g OGTT. We believe that a better understanding of the pathophysiological aspects of alteration in glucose metabolism, related to betamethasone action, could be helpful in everyday clinical practice. Furthermore, it would improve the quality of the management of pregnancies at risk, requiring BM administration, by confronting, on time, any possible side effects that may be encountered from hyperglycemia in an already at-risk pregnancy.

## Figures and Tables

**Figure 1 healthcare-08-00040-f001:**
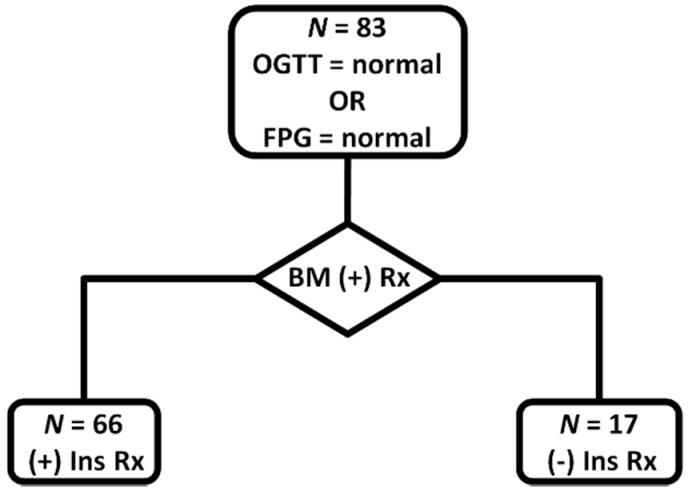
Flowchart of the glycemic management for the pregnant women that were studied; OGTT: oral glucose tolerance test; FPG: fasting plasma glucose; BM (+) Rx: betamethasone treatment; (+) Ins Rx: insulin treatment; (-) Ins Rx: no insulin treatment needed.

**Figure 2 healthcare-08-00040-f002:**
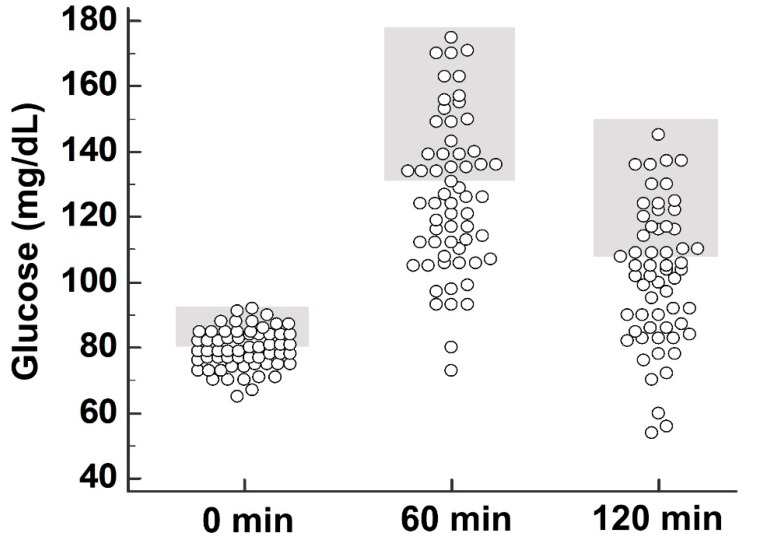
Results of the oral glucose tolerance test with 75 g of glucose. Glucose values at 0 min, 60 min, and 120 min. Marginal values per HAPO category 3 and higher are shown in grey.

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
