# Peer review of "Glycemia after Betamethasone in Pregnant Women without Diabetes—Impact of Marginal Values in the 75-g OGTT"

_healthcare, 2020, doi:10.3390/healthcare8010040_

Round 1

Reviewer 1 Report

In an observational study, 83 women with normal glucose metabolism were recruited after receiving betamethasone (BM) from their hospitalization for at-risk pregnancy. The authors found increased glucose levels following BM and a large proportion of pregnant women requiring insulin administration for a short period of time.
This was associated with the dosing of BM but not with other key parameters such as gestational age, weight gain, or hidramnios.

Paper is well-written and organized. Results are interesting but expected.

Methods. A description of what is meant by normal glucose metabolism and the timing relating glucose concentrations before and after BM administration is important to better understanding all the process. A flowchart could help.

Author Response

Addressing the first reviewer’s comments, we changed the title in the article in order to be more specific regarding our study’s findings. We believe that the most important and novel finding is the impact of marginal values in the OGTT, in a statistically significant relation vis-à-vis insulin administration, regarding the need for betamethasone (BM) administration. This finding could possibly indicate a mild/latent glucose homeostasis disorder, revealed by the administration of BM. The need for insulin administration after BM in pregnant women without known diabetes, has been assessed in few studies. Our findings, regarding the marginal values in the OGTT, provide another perspective in the management of hyperglycemia after BM. Furthermore, we added more information in the Materials and Methods section, regarding normal glucose metabolism definition and our study’s inclusion and exclusion criteria, along with timing relating expected hyperglycemia based on the literature.

Reviewer 2 Report

The authors do not repeat known studies. This is an original study. The authors described the required volume of patients. However, the article would be more interesting if the authors describe the population of patients who entered the study. Are these Europeans, Latinos or Chinese? Could the authors inсlude a table of the distribution of patients into groups in the materials and methods section. This will facilitate the perception of the material. The authors must supplement the article with a conclusion.

Author Response

Regarding the second reviewer’s comments, we clarified in the Materials and Methods section, that all the women included in the study were Greeks/Caucasians homogeneity reasons. For this reason, we did not include a table of patients’ distribution. We also added a Conclusions sector in the text, as the reviewer rightly recommended.

Round 2

Reviewer 1 Report

All my concerns have been adequately addressed